# Cross-Reactivity of N6AMT1 Antibodies with Aurora Kinase A: An Example of Antibody-Specific Non-Specificity

**DOI:** 10.3390/antib13020033

**Published:** 2024-04-22

**Authors:** Baiba Brūmele, Evgeniia Serova, Aleksandra Lupp, Mihkel Suija, Margit Mutso, Reet Kurg

**Affiliations:** Institute of Technology, University of Tartu, 50411 Tartu, Estonia; baiba.brumele@ut.ee (B.B.); evgeniia.serova@ut.ee (E.S.); aleksandra.lupp@ut.ee (A.L.); mihkel.suija@ut.ee (M.S.); reet.kurg@ut.ee (R.K.)

**Keywords:** methyltransferase, N6AMT1, AURKA, antibodies, cross-reactivity

## Abstract

Primary antibodies are one of the main tools used in molecular biology research. However, the often-occurring cross-reactivity of primary antibodies complicates accurate data analysis. Our results show that three commercial polyclonal antibodies raised against N-6 adenine-specific DNA methyltransferase 1 (N6AMT1) strongly cross-react with endogenous and recombinant mitosis-associated protein Aurora kinase A (AURKA). The cross-reactivity was verified through immunofluorescence, immunoblot, and immunoprecipitation assays combined with mass spectrometry. N6AMT1 and AURKA are evolutionarily conserved proteins that are vital for cellular processes. Both proteins share the motif ENNPEE, which is unique to only these two proteins. We suggest that N6AMT1 antibodies recognise this motif in N6AMT1 and AURKA proteins and exhibit an example of “specific” non-specificity. This serves as an example of the importance of controls and critical data interpretation in molecular biology research.

## 1. Introduction

Primary antibodies are widely used as tools in life science research. Antibodies have enabled life scientists to detect specific targets through various molecular biology methods, such as immunoblot, immunofluorescence, immunohistochemistry, immunoprecipitation, etc. The cross-reactivity of antibodies used for research purposes is a well-known phenomenon that often causes issues in correctly interpreting scientific results [1]. In 2008, a study claimed that only half of the 6000 antibodies tested recognised their intended targets [2]. It has even been suggested that poor antibodies are a significant reason why it has not been possible to replicate the scientific results of many landmark preclinical studies [3]. The scarcity of reliable antibodies is particularly problematic when investigating proteins.

The N-6 adenine-specific DNA methyltransferase 1 (N6AMT1) is an evolutionarily conserved protein [4,5,6], which, together with the co-factor TRMT112, functions as a protein methyltransferase [7,8]. N6AMT1 is suggested to be involved in the regulation of vital cell processes, such as the cell cycle, cell proliferation, cell division, apoptotic processes, and programmed cell death [8,9,10,11,12]. There is an ongoing debate and conflicting results on whether N6AMT1 is involved in DNA methylation, specifically 6 mA modification [13,14,15]. 

During our research to elucidate the functions of N6AMT1, we have encountered issues related to antibody cross-reactivity. Currently, 22 different N6AMT1 antibodies are publicly available, and in this study, we analysed 6 different commercially available antibodies. We found that two of them failed to recognise endogenous or recombinant N6AMT1, one recognised both but had a very strong background, and three polyclonal antibodies strongly cross-reacted with the mitosis-related protein Aurora kinase A in both linear and folded protein structures. Aurora kinase A (AURKA) is a serine/threonine kinase that is crucial in mitotic spindle assembly and cell division. Its expression level and activity peak during the G2/M phase of the cell cycle, and it is essential for the proper progression of cell division [16,17]. In silico analysis revealed that N6AMT1 and AURKA share the protein motif ENNPEE, which is unique to only these two proteins. We suggest that N6AMT1 antibodies recognise this motif in N6AMT1 and AURKA proteins and exhibit an example of “specific” non-specificity. To our knowledge, prior studies have not detected N6AMT1 antibody cross-reactivity with Aurora kinase A. 

## 2. Materials and Methods

### 2.1. Cell Culture

Human osteosarcoma cells (U2OS) (American Type Culture Collection, Manassas, VA, USA) and N6AMT1 knockout cell line ΔN6AMT1#1 [12] were grown in complete culture media (Iscove’s Modified Dulbecco’s Medium (IMDM)) supplemented with 10% foetal calf serum (FCS) (Gibco, Thermo Fisher Scientific, Waltham, MA, USA), 100 U/mL penicillin, and 100 µg/mL streptomycin. The cells were incubated at 37 °C in a 5% CO_2_ environment.

### 2.2. Construction of Cell Lines

The generation of U2OS N6AMT1-knockout cell line ΔN6AMT1#1 is described in [12]. A stable cell line of ΔN6AMT1#1 expressing N6AMT1-EGFP fusion protein ΔN6AMT1#1 comp. was generated similarly as described in Õunap et al. [18]. pQM-N6AMT1-EGFP and pBabe-Puro plasmids were linearised, ligated to form dimers, and transfected to ΔN6AMT1#1 cells using the electroporation method described previously by Brūmele et al. [19]. A duration of 24 h after transfection, puromycin was added to the media at a final concentration of 5 μg/mL. Colonies were selected two weeks after transfection, and the expression of N6AMT1-EGFP protein was analysed by immunoblotting. 

### 2.3. Plasmids

The open reading frame for Aurora kinase A (UniProt accession Q5QPD1) was amplified from the cDNA obtained from U2OS cells and cloned into pEGFP-C1 and pEGFP-N1 plasmids in a manner similar to that described in [20], using the primers F 5′ACGACTCGAGGTATGGACCGATCTAAAGAA and R 5′AGCAGGATCCCTAAGACTGTTTGCTAGC, and F 5′ACGAGGTACCATGGACCGATCTAAAGAA and R 5′AGCAGGATCCCCAGACTGTITGCTAGCTGA, respectively. Plasmid sequences were controlled by Sanger sequencing.

### 2.4. Immunofluorescence Microscopy Analysis

Cells were seeded on coverslips in 24-well plates and, after 24 h, they were washed with PBS, fixed with 4% paraformaldehyde for 10 min, and permeabilised with 0.2% Triton-X-100 for 2 min at RT. Samples were blocked with 3% Bovine Serum Albumin (BSA)/PBS solution for 1 h at RT and stained with anti-N6AMT1 (1:100, CQA1550, Cohesion Biosciences, London, UK), anti-N6AMT1 (1:100, HPA059242, Atlas antibodies, Bromma, Sweden), anti-N6AMT1 (1:100, 16211-1-AP, Proteintech, Rosemont, IL, USA), anti-N6AMT1 (1:100, PA5-121076, Invitrogen, Waltham, MA, USA), anti-N6AMT1 (1:100, ARP45845_P050, Aviva Systems Biology, San Diego, CA, USA), anti-α-tubulin (1:5000, T5168, Sigma-Aldrich, St. Louis, MO, USA), and pericentrin (1:1000, ab28144, Abcam, Cambridge, UK) diluted in 3% BSA/PBS solution, followed by three washes with PBS, and incubation with secondary anti-mouse and anti-rabbit antibodies conjugated to Alexa Fluor 568 or 488 (1:1000, Invitrogen, Carlsbad, CA, USA) diluted in 3% BSA/PBS solution. The nuclei were counterstained with DAPI. The samples were analysed using a Zeiss LSM 900 confocal microscope and Zen blue 3.1 software (Carl Zeiss AG, Oberkochen, Germany).

### 2.5. Immunoblot Analysis

The cells were collected and lysed in 2× Laemmli buffer supplemented with 100 mM DTT and denatured at 100 °C for 10 min. Immunoblotting was performed as previously described [21]. Proteins were detected using anti-N6AMT1 (1:1000, CQA1550, Cohesion Biosciences, London, UK), anti-N6AMT1 (1:2000, HPA059242, Atlas antibodies, Bromma, Sweden), anti-N6AMT1 (1:500, sc-517120, Santa Cruz Biotechnology, Dallas, TX, USA), anti-N6AMT1 (1:1000, 16211-1-AP, Proteintech, Rosemont, IL, USA), anti-N6AMT1 (1:1000, PA5-121076, Invitrogen, Waltham, MA, USA), anti-N6AMT1 (1:1000, ARP45845_P050, Aviva Systems Biology, San Diego, CA, USA), anti-GAPDH (1:10,000, sc-32233, Santa Cruz Biotechnology, Dallas, TX, USA), anti-Aurora kinase A (1:500, sc-56881, Santa Cruz Biotechnology, Dallas, TX, USA), and anti-EGFP (1:10,000, Institute of Technology; University of Tartu, Tartu, Estonia). Goat anti-rabbit-HRP (1:10,000, 31460, Invitrogen, Carlsbad, CA, USA) and goat anti-mouse-HRP (1:10,000, 31430, Invitrogen, Carlsbad, CA, USA) were used as secondary antibodies.

### 2.6. Cell Cycle Analysis

The mitotic cell population was obtained after incubation with complete culture media supplemented with 50 ng/mL nocodazole for 10 h, followed by mitotic shaking. Only the mitotic cells were collected. For cell cycle progression out of mitosis, 2 × 10^5^ mitotic cells per 60 mm plate were seeded in complete culture media. Cells were incubated for 0 h, 1 h, 2 h, 3 h, 6 h, 9 h, 12 h, 15 h, 18 h, 21 h, 24 h, 27 h, 30 h, and 33 h in complete media and collected for immunoblot assay.

### 2.7. Immunoprecipitation

Unsynchronised cells were collected 18 h after seeding them on 100 mm plates and lysed in 500 μL IP buffer (10 mM Tris, pH 7.5, 100 mM KCl, 2 mM MgCl_2_, 1 mM dithiothreitol, 0.5% NP-40, and protease inhibitors) on ice for 30 min. The supernatants were collected after centrifugation at 13,000 rpm for 10 min at 4 °C. Protein G beads conjugated with anti-N6AMT1 (1:200, CQA1550, Cohesion Biosciences, London, UK) or anti-TRMT112 (1:100, sc-398481, Santa Cruz Biotechnology, Dallas, TX, USA) were added and incubated at 4 °C overnight. The beads were washed three times with IP buffer. Magnetic beads with attached proteins were lysed in 2× Laemmli buffer containing 100 mM DTT and heated to 100 °C for 10 min. 

### 2.8. Mass Spectrometry

To identify the unknown protein via mass spectrometry, approximately 40–55 kDa proteins were excised from the polyacrylamide gel after SDS gel electrophoresis, which was performed as described above. The cut-out gel piece was transferred to LoBind (Eppendorf) tubes and sliced into 1 mm^3^ pieces. A measure of 1 mL of 1:1 acetonitrile/50 mM HEPES pH 8.5 was added to the gel pieces and vortexed for 30 min, after which the supernatant was discarded. Next, 600 μL acetonitrile was added, followed by 5 min incubation, after which the supernatant was removed. The proteins were reduced and alkylated by adding 10 mM TCEP and 40 mM CAA, followed by vortexing and 5 min incubation at 70 °C, and the supernatant was discarded. Next, 600 μL acetonitrile was added, followed by 10 min incubation; the supernatant was removed and the gel pieces were dried completely. Next, 200 μL of 10 ng/μL proteomics-grade protease (TeyP) in ice-cold 50 mM HEPES (pH 8.5) was added to the gel pieces, followed by 2 h incubation on ice. The sample was incubated overnight at 37 °C, followed by sonication in a sonication bath the following day. Next, 400 μL of 1:2 ratio of 5% formaldehyde/acetonitrile was added, followed by vortexing for 15 min. The extract was then transferred to a fresh LoBind tube and vacuum-centrifuged to 10–20 μL to remove acetonitrile and formaldehyde. Next, 200 μL of 1% TFA was added, followed by sonication for 3 min. Next, the sample was vortexed, and peptides were cleaned on a C18 (3M Empore) StageTip, suspended to a volume of 22.1 μL of 0.5% TFA with 25 μg of iRT mix, then sonicated and vortexed again. The sample was loaded onto a Protein LoBind 384-well plate and analysed using Thermo Fisher Scientific Q Exactive HF. Protein identification and quantification were performed using the MaxQuant software package (version 1.1.1.36).

### 2.9. Sequence Analysis

The BlastP sequence alignment tool with the default parameters was used to align AURKA (UniProt accession O14965) and N6AMT1 (UniProt accession Q9Y5N5) [22]. Full AURKA and N6AMT1 sequences were aligned using ClustalΩ ver. 1.2.2 [21] with default options in SeaView ver. 4.7 [23]. The motifs ENNPPE and EKVDL were aligned separately with their respective N6AMT1 flanking sequences against the partial sequence of AURKA with ClustalΩ ver. 1.2.4 [21] using the online service of EMBL-EBI “https://www.ebi.ac.uk/jdispatcher/msa/clustalo (accessed on 15 January 2024)”. To assess the potential of ENNPEE and EKVDL motifs as possible antibody epitopes, we analysed the respective motifs on the protein structure of both N6AMT11 and AURKA using the crystal structure of N6AMT1-TRMT112 complex described in Li et al. 2019 [14] (PDB accession 6KMR) and the AlphaFold predicted structure of AURKA (PDB accession AF_AFO14965F1) in ChimeraX ver. 1.4 [24]. To further screen for proteins that shared either of the two motifs, the Genomenet Motif finder tool was used against the KEGG Genes database and was set to search only for human proteins “https://www.genome.jp/ (accessed on 15 January 2024)”.

## 3. Results

### 3.1. Antibodies against N6AMT1 Protein

The availability of specific antibodies is of crucial importance when studying proteins. We have identified twenty-two N6AMT1 antibodies currently available on the market (Table 1). 

Several research groups have used them for immunoblotting [19,25], immunohistochemistry [8,9,28], and immunoprecipitation [8,11,29] assays (Table 1). To analyse the specificity of N6AMT1 antibodies, we selected six N6AMT1 antibodies used in previous research (HPA059242, CQA1550, 16211-1-AP, ARP45845_P050, PA5-121076, and sc-517120) (Table 1, nr I–VI) for further analysis. Five antibodies were rabbit polyclonals, and only one was mouse monoclonal. These antibodies were raised against different N6AMT1 immunogens, ranging from small synthetic peptides to full-length recombinant proteins. The most cited antibody, #27630 [8,10,11,28], was generated in-house and is not commercially available (Table 1, nr VII). One of the cited antibodies (sc-83304) [29] has since been discontinued (Table 1, nr VIII).

### 3.2. N6AMT1 Antibody Recognises Protein in the Centrosomes during Mitosis

We used six N6AMT1 antibodies—antibody I (CQA1550), antibody II (HPA059242), antibody III (16211-1-AP), antibody IV (PA5-121976), antibody V (ARP45845), and antibody VI (sc-517120)—to study the cellular localisation of N6AMT1 by immunofluorescence microscopy in U2OS cells. In all cases, the signal was detected in both the cytoplasm and nucleus in U2OS interphase cells (Figure 1A; Appendix A). A similar pattern was observed using the same set of antibodies for U2OS-ΔN6AMT1#1 cell line, where N6AMT1 was knocked out using the CRISPR/Cas9 system [12] (Figure 1B; Appendix A). In mitotic cells, antibodies I and II showed strong signals at the centrosomes (Figure 2). For antibody III, similar but much weaker signals were detected at the centrosomes, whereas for antibody IV, no centrosome staining was observed. The immunofluorescence signals for antibodies V (ARP45845) and VI (sc-517120) were below the detection levels. Again, similarly to interphase cells (Figure 1A,B; Appendix A), mitotic U2OS and U2OS-ΔN6AMT1#1 cells showed similar immunofluorescence signal patterns (Figure 2A,B; Appendix A). All cells were co-stained with α-tubulin as a control and DAPI to confirm the cell cycle stage of the cell (Figure 1 and Figure 2). α-tubulin and β-tubulin dimers polymerise to create microtubules which organise into a web-like structure in the interphase and form the mitotic spindle in mitosis [30]. Pericentrin localises at the centrosomes and is involved in the mitotic spindle organisation [31]. Therefore, our study used it as a centrosome marker (Appendix A). As shown in Appendix A, pericentrin co-localized with the N6AMT1 signal in U2OS as well as in U2OS-ΔN6AMT1#1 cells, confirming that N6AMT1 antibodies recognise a protein that localises at the centrosomes during mitosis.

During these experiments, we started to doubt the specificity of commercially available N6AMT1 antibodies because U2OS-ΔN6AMT1#1 cells showed an apparent phenotype that distinguished them from U2OS cells [12] and is consistent with other studies [8,9,10,11]. Our subsequent experiments were designed to analyse whether N6AMT1 is these antibodies’ main and only target.

### 3.3. N6AMT1 Antibody Recognises Multiple Proteins in Immunoblot Analysis

Next, N6AMT1 antibodies I, II, III, IV, V, and VI were validated by immunoblotting analysis using U2OS cells and the N6AMT1 knockout cell line ΔN6AMT1#1. As a positive control, a compensatory cell line, ΔN6AMT1#1 comp., expressing a recombinant N6AMT1-EGFP fusion protein with a size of 55 kDa, was used.

Antibody I detected two bands with molecular masses of 23 kDa and 45–50 kDa in U2OS cells (Figure 3A, lane 1). The 23 kDa band corresponds to the reported size of the N6AMT1 protein. In the ΔN6AMT1#1 cell line, there was no visible signal at the 23 kDa, while the band at 45–50 kDa remained unchanged (Figure 3A, lane 2). In the case of the ΔN6AMT1#1 comp., a strong band at 55 kDa, corresponding to the recombinant N6AMT1-EGFP fusion protein, was observed, confirming that the antibody indeed recognises the N6AMT1 protein (Figure 3A, lane 3). An antibody against GAPDH was used as a loading control. Antibodies II (Figure 3B) and III (Figure 3C) displayed very similar results to antibody I (Figure 3A), with a visible band at 23 kDa (Figure 3B,C; lanes 1), no signal at the 23 kDa in ΔN6AMT1#1 (Figure 3B,C; lanes 2), and a strong band at 55 kDa in the ΔN6AMT1#1 comp. cell line (Figure 3B,C; lanes 3). In the case of antibody IV (Figure 3D), multiple additional bands of various sizes were observed; nonetheless, a band at 23 kDa was detected (Figure 3D, lane 1), and no signal was detected at 23 kDa in the case of ΔN6AMT1#1 (Figure 3D, lane 2), while the antibody also recognised a band at 55 kDa in the ΔN6AMT1#1 comp cell line (Figure 3D, lane 3), confirming that this antibody recognises N6AMT1 as well despite the additional non-specific bands. In the case of antibody V, no signal was observed at 23 kDa (Figure 3E, lane 1) in U2OS cells; meanwhile, in the case of antibody VI, a signal observed at 23 kDa in U2OS cells was also detected in ΔN6AMT1#1 cells (Figure 3F; lanes 1,2). Furthermore, no signal corresponding to the N6AMT1-EGFP size was observed for antibodies V and VI (Figure 3E,F; lanes 3), suggesting that these antibodies do not recognise N6AMT1 protein in U2OS cells in immunoblot analysis. 

Our results showed that the N6AMT1 antibodies I, II, III, and IV recognised N6AMT1 protein in the immunoblot assays. In contrast, V and VI did not, as endogenous and recombinant N6AMT1-EGFP could not be detected with either of these antibodies. All the antibodies examined recognised some additional protein(s) besides N6AMT1. 

### 3.4. N6AMT1 Antibody Recognises a Mitosis-Associated Protein in Immunoblot Analysis

In previous research, N6AMT1 has been linked to cell cycle regulation [8,9,10,11,12]; therefore, we designed an experiment to determine N6AMT1 levels during different cell cycle stages (Figure 4). Mitotic cells were collected from U2OS (Figure 4A) and ΔN6AMT1#1 (Figure 4B) cell lines, released from the mitotic block and collected at designated time points from 0 h to 33 h to follow the progression of the cell cycle. As shown in Figure 4, the 23 kDa band corresponding to N6AMT1 was detected in U2OS cells (Figure 4A) but missing in ΔN6AMT1#1 cells (Figure 4B). However, an additional 45–50 kDa band exhibited a cell cycle dependency in U2OS and ΔN6AMT1#1 cells. Immunoblot staining with antibody I showed a strong signal at 0 h and 1 h (Figure 4A,B; lanes 1–2), corresponding to mitotic cells. At 2 h post-mitotic release, the signal at 45–50 kDa sharply decreased (Figure 4A,B; lanes 3–7) and started to rise again 15 h after seeding of cells (Figure 4A,B; lanes 8–11), which corresponds to cells reaching the G2 phase as a complete U2OS cell cycle is reported to be approximately 18–20 h [32]. These data suggest that the N6AMT1 polyclonal antibody I (CQA1550) recognises a protein with high expression during mitosis. 

### 3.5. N6AMT1 Antibody Immunoprecipitates Aurora Kinase A

The immunofluorescence and immunoblot assays showed that at least three different N6AMT1 antibodies recognised a mitosis-specific protein with a high affinity. Therefore, we aimed to identify the protein that is, in addition to N6AMT1, recognised by N6AMT1-specific antibodies.

Mitotic U2OS cells were collected and lysed for the immunoprecipitation assay using the N6AMT1 antibody I. As a control, an antibody against the known N6AMT1 interactor TRMT112 [22] was used (Figure 4C). The immunoprecipitated samples were analysed by immunoblotting using antibody I. This N6AMT1 antibody was able to precipitate N6AMT1 as well as an unknown protein with a size of approximately 45–50 kDa (Figure 4C; lane 3), while the TRMT112 antibody did not immunoprecipitate any of them (Figure 4C; lane 2). To identify the unknown protein, proteins moving at 40–55 kDa were excised from the polyacrylamide gel after SDS gel electrophoresis and subjected to mass spectrometry analysis (Figure 4D). The proteins detected by proteomics are listed in Appendix A. The most likely candidate for the unknown protein with a molecular weight of approximately 45–50 kDa was Aurora kinase A (AURKA, ARK1, Aurora-A kinase, Aurora-related kinase 1), as it is highly expressed during mitosis and localises to the mitotic spindle poles [33].

### 3.6. Multiple Commercial Antibodies Raised against N6AMT1 Recognise Aurora Kinase A

To verify whether the N6AMT1 antibody recognises Aurora kinase A, plasmids expressing recombinant Aurora kinase A protein fused with EGFP were constructed. Plasmid pEGFP-AURKA expressed Aurora kinase A with EGFP at the N-terminus, and the pAURKA-EGFP plasmid expressed Aurora kinase A with EGFP at its C-terminus. The EGFP expression plasmids pEGFP_C1 and pEGFP-N1 without insertion were used as controls. The EGFP-AURKA (Figure 5A, lane 1) and AURKA-EGFP (Figure 5A, lane 3) proteins were expressed in U2OS cells and detected with EGFP and AURKA-specific antibodies as 72 kDa and 85 kDa fusion proteins, respectively. Next, the same samples were analysed with six N6AMT1 antibodies used in previous experiments: I, II, III, IV, V, and VI. As shown in Figure 5B–D, antibodies I, II, and III recognise recombinant Aurora kinase A-EGFP fusion proteins (Figure 5B–D; lanes 1, 3). The antibody IV showed many bands with different sizes, including a similar size to the AURKA-EGFP and EGFP-AURKA recombinant proteins; therefore, it was not possible to conclude whether antibody IV recognised AURKA (Figure 5E). Antibodies V and VI did not recognise recombinant or endogenous AURKA (Figure 5F,G). These experiments showed that three commercial N6AMT1 antibodies react specifically with Aurora kinase A.

### 3.7. N6AMT1 and Aurora Kinase A Share Similar Motifs

Antibody cross-reactivity usually occurs due to high similarities in protein sequences [34]. We used the BlastP sequence alignment tool with the default parameters [22] to screen for potential protein sequence similarities or motifs between AURKA (Uniprot accession O14965) and N6AMT1 (Uniprot accession Q9Y5N5). The alignment revealed two motifs present in both N6AMT1 and Aurora kinase A proteins (Figure 6). Figure 6A shows N6AMT1 immunogens used for raising antibodies and the location of these motifs within the N6AMT1 protein. Motif ENNPEE was found in N6AMT1 at position 176–181 and Aurora kinase A at position 107–112 (Figure 6B and Appendix A), while motif EKVDL was found in N6AMT1 at aa position 114–118 and in AURKA 308–312 (Figure 6E). The Genomenet Motif finder tool was used against the KEGG Genes database to analyse further how common these motifs are and was set to only look for human proteins “https://www.genome.jp/ (accessed on 15 January 2024)”. The motif screening revealed that ENNPEE is rare and is shared only by two proteins: N6AMT1 and AURKA. The motif EKVDL is more common, and 26 different human proteins contain this motif. In addition to N6AMT1 and Aurora kinase A, the EKVDL motif is present in Aurora kinases B and C. 

The structural analysis of AURKA and N6AMT1-TRMT112 complexes revealed that the ENNPEE motif is located in the freely accessible regions in both AURKA and N6AMT1 (Figure 6C,D). In contrast, motif EKVDL is much less accessible and hidden in the secondary structure of both AURKA and N6AMT1 (Figure 6F,G). Motif ENNPEE is also present in all N6AMT1 immunogens used for raising antibodies I, II, and III (Figure 6A), which specifically recognised the AURKA protein in Western blot analysis (Figure 5B–D), suggesting that polyclonal antibodies raised against N6AMT1 recognise preferentially this motif. Therefore, our data indicate that N6AMT1 antibodies I,II, and III analysed in the current study most likely recognise the ENNPEE motif in N6AMT1 and AURKA proteins.

## 4. Discussion

Antibody cross-reactivity is a well-known issue in biomedical research and could be one of the reasons for poor data reproducibility in academic and industrial research fields [1,3,35]. In the current study, we analysed the cross-reactivity of N6AMT1 antibodies using classical biomedical laboratory methods, including immunoblotting, immunofluorescence, and immunoprecipitation assays. Immunoblotting analysis revealed that all six analysed N6AMT1 antibodies showed significant non-specific background staining and only four antibodies recognised N6AMT1 with expected molecular weight. Interestingly, we noticed that three N6AMT1 antibodies had similarly sized non-specific bands, which were confirmed to be Aurora kinase A. 

Antibody cross-reactivity is often caused due to similarities between protein sequences or secondary structures. In the case of N6AMT1 and Aurora kinase A, the only similarities in the aa sequence we found were EKVDL and ENNPEE motifs. The ENNPEE motif is shared in humans only between N6AMT1 and AURKA. Meanwhile, EKVDL is found in multiple proteins; among others, it is also shared by Aurora kinases B and C. Currently, no information is available regarding the biological functions of either of these motifs. Our findings demonstrate that three N6AMT1 antibodies, I (CQA1550), II (HPA059242), and III (16211-1-AP), recognised AURKA in both linear and folded protein structures, as they cross-react in immunoblot as well as immunoprecipitation and immunofluorescence assays. Different companies produce these three antibodies, and different immunogens have been used for immunisation. All three antibodies contain the ENNPEE motif in their antigen, but only I and III contain the EKVDL motif. This might indicate that ENNPEE could be the motif responsible for the cross-reactivity.

Meanwhile, N6AMT1 antibody IV was also raised against the N-terminal part of the protein from 115–214 and included a full-length ENNPEE motif and partial EKVDL motif. This antibody produced multiple unspecific bands in immunoblot but did not recognise AURKA recombinant protein in the immunoblot assay with high affinity. Nonetheless, we suggest that antibodies I, II, and III most likely recognise the ENNPEE motif present both in N6AMT1 and AURKA proteins.

In immunoblot assays, antibody cross-reactivity usually does not cause data interpretation issues, assuming that proteins have different molecular weights, such as endogenous N6AMT1 and AURKA. However, it can be an issue if the non-specific signal is the same size as expected for the target protein. For example, when performing immunoblot to validate a siRNA knockdown, the siRNA target protein has the same molecular weight as a protein of interest. Furthermore, antibody cross-reactivity can cause significant misinterpretation of data when someone wants to study the protein interactions through immunoprecipitation assays. Mouse or rabbit IgGs are commonly used as controls in these assays; however, in cases where the primary antibody used cross-reacts with proteins other than the intended target, it might not provide sufficient validation. This would be the case with some of the analysed N6AMT1 antibodies, as they successfully immunoprecipitated AURKA.

N6AMT1 is reported to be localised in centrosomes when analysed with immunofluorescence microscopy. Indeed, we detected a similar pattern with three different N6AMT1 antibodies. Unexpectedly, the localisation did not change in the N6AMT1 knockout cells, which raises the question of the actual localisation of N6AMT1 in the cells. The same three N6AMT1 antibodies that recognised AURKA in immunoblot assays localised N6AMT1 at the centrosomes. Our data suggest that the centrosomal pattern of N6AMT1 detected by N6AMT1 antibodies is a signal of AURKA rather than N6AMT1. At the same time, N6AMT1 localisation at the centrosomes cannot be excluded entirely, as it may be hidden under the stronger AURKA staining.

It is well known that private companies in the industry often struggle to reproduce results obtained by academic researchers [36,37,38]. The cross-reactivity of N6AMT1 with AURKA is an excellent example of the importance of antibody validation and controls during the research process. Insufficient controls may lead to misinterpretation of data and contribute to the overall problem of data reproducibility. This leads to a vast waste of resources in the biomedical research field, significantly hindering scientific discovery [38]. 

## Figures and Tables

**Figure 1 antibodies-13-00033-f001:**
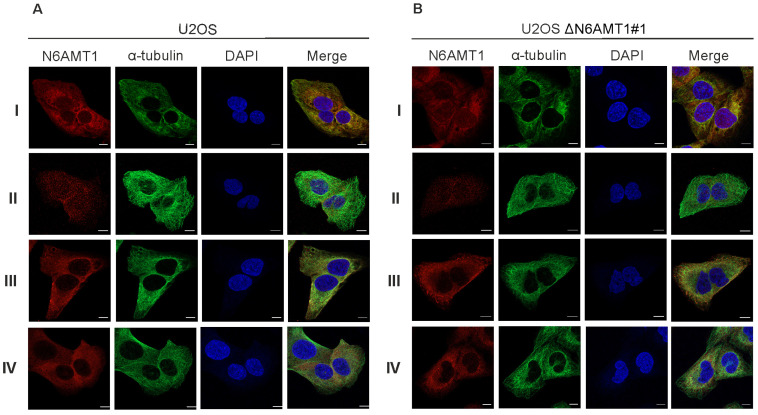
The target of the N6AMT1 antibody localises in the cytoplasm during interphase: (**A**) U2OS cells; (**B**) N6AMT1 knockout cells ΔN6AMT1#1, processed for immunofluorescence with primary antibodies specific to N6AMT1 (red): I (CQA1550), II (HPA059242), III (6211-1-AP), and IV (PA5-121076); α-tubulin (green) and secondary antibodies conjugated with Alexa-568 and Alexa 488. The cells were then counterstained with DAPI (blue) for DNA labelling. Images were captured using a Zeiss LSM 900 confocal microscope at 63× magnification. Scale bar, 10 μm.

**Figure 2 antibodies-13-00033-f002:**
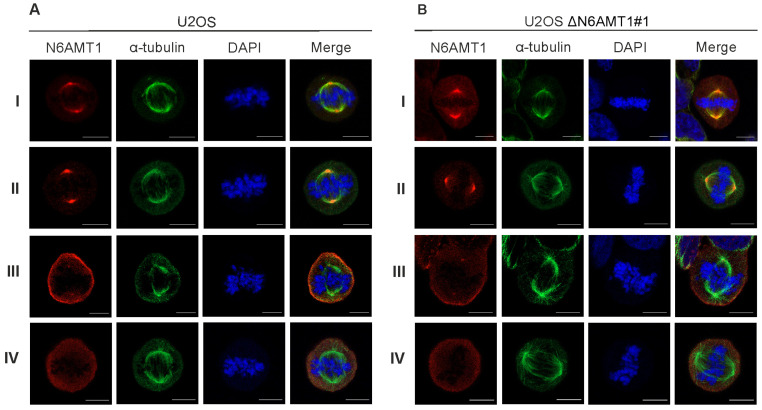
The target of the N6AMT1 antibody localises at the centrosomes during mitosis. (**A**) U2OS cells, mitosis; (**B**) N6AMT1 knockout cells ΔN6AMT1#1, mitosis, processed for immunofluorescence with primary antibodies specific to N6AMT1 (red): I (CQA1550), II (HPA059242), III (6211-1-AP), and IV (PA5-121076); α-tubulin (green) and secondary antibodies conjugated with Alexa-568 and Alexa 488. The cells were then counterstained with DAPI (blue) for DNA labelling. Images were captured using a Zeiss LSM 900 confocal microscope at 63× magnification. Scale bar, 10 μm.

**Figure 3 antibodies-13-00033-f003:**
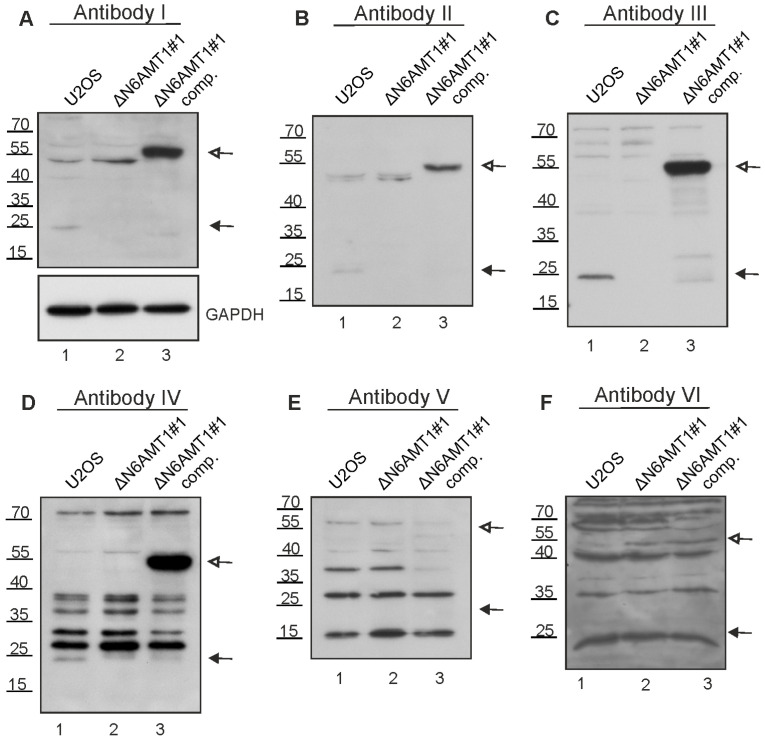
Immunoblotting with N6AMT1 antibodies. Lane 1—U2OS cell line; lane 2—N6AMT1 knockout cell line ΔN6AMT1#1; lane 3—N6AMT1 knockout compensatory cell line ΔN6AMT1# comp. N6AMT1 endogenous level in U2OS cells corresponds to 23 kDa (lane 1); no band at 23 kDa should be observed in knockout cell line ΔN6AMT1#1 (lane 2); recombinant N6AMT1-EGFP size corresponds to 55 kDa (lane 3). Immunoblot was performed with N6AMT1 antibody: (**A**) I (CQA1550), (**B**) II (HPA059242), (**C**) III (16211-1-AP), (**D**) IV (PA5-121076), and (**E**) V (ARP45845_P050), and (**F**) VI (Sc-517120). The band corresponding to endogenous N6AMT1 at 23 kDa is marked with a filled-in arrow, and recombinant N6AMT1-EGFP at 55 kDa is marked with an empty arrow.

**Figure 4 antibodies-13-00033-f004:**
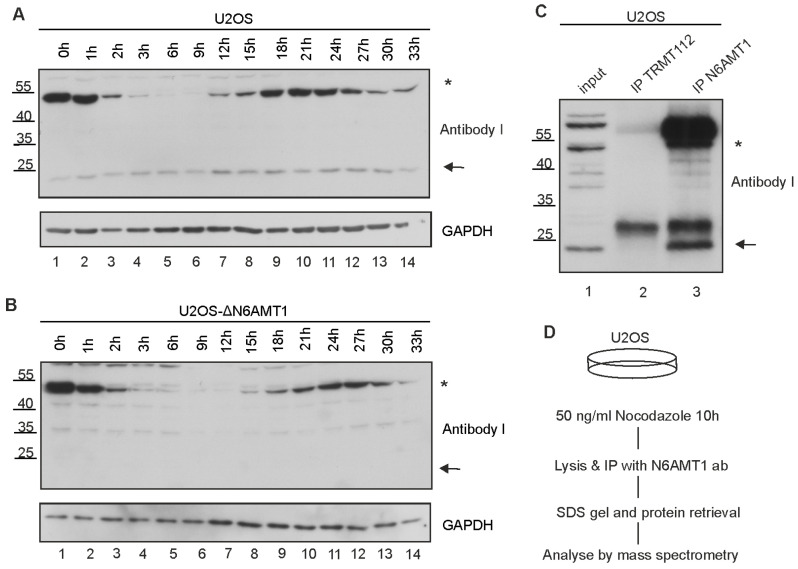
N6AMT1 polyclonal antibody recognises an unknown protein with a size of 45–50 kDa in U2OS cells. Mitotic cells of (**A**) U2OS and (**B**) ΔN6AMT1#1 were seeded, released from the mitotic block and collected at 0 h–33 h time points. Samples were analysed using N6AMT1 antibody I (CQA1550) and GAPDH. (**C**) Immunoprecipitation analysis in mitotic U2OS cells. Input (lane 1), TRMT112 (lane 2), and N6AMT1 (lane 3) antibodies were used for co-immunoprecipitation. Samples were analysed by immunoblotting with antibodies against N6AMT1. The band corresponding to N6AMT1 at 23 kDa is marked with an arrow, and the additional mitosis-associated band at 45–50 kDa is marked with an asterisk. (**D**) Experimental scheme of mass spectrometry analysis to identify the unknown protein that the N6AMT1 antibody I (CQA1550) recognises at 45–50 kDa.

**Figure 5 antibodies-13-00033-f005:**
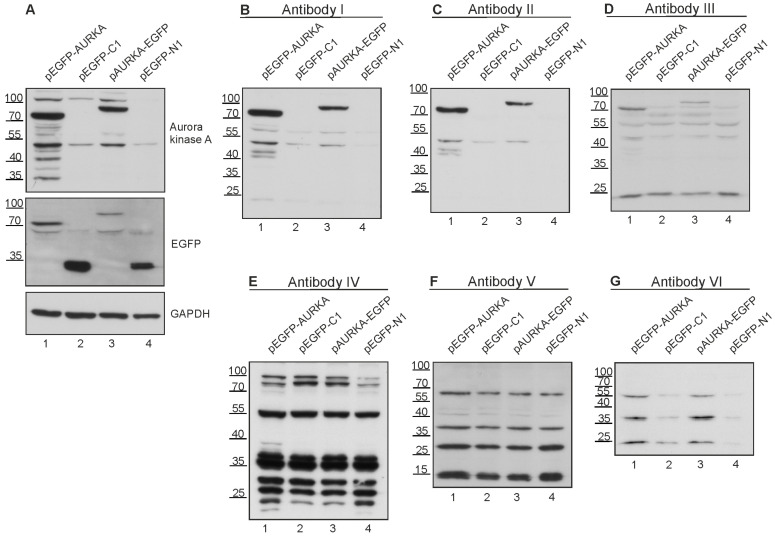
Multiple N6AMT1 polyclonal antibodies recognise Aurora kinase A in U2OS cells. (**A**) U2OS cells were transfected with recombinant Aurora kinase A-EGFP fusion protein in which EGFP was fused in the N (EGFP-AURKA, 77 kDa, lane 1) or C terminus (AURKA-EGFP, 85 kDa, lane 3). EGFP plasmid vectors were used as controls (lanes 2 and 4). Samples were analysed by immunoblotting using antibodies against Aurora kinase A, EGFP, and GAPDH. (**B**–**G**) Samples were analysed using N6AMT1 antibodies I (CQA1550), II (HPA059242), III (6211-1-AP), IV (PA5-121076), V (ARP45845_P050), and VI (sc-517120).

**Figure 6 antibodies-13-00033-f006:**
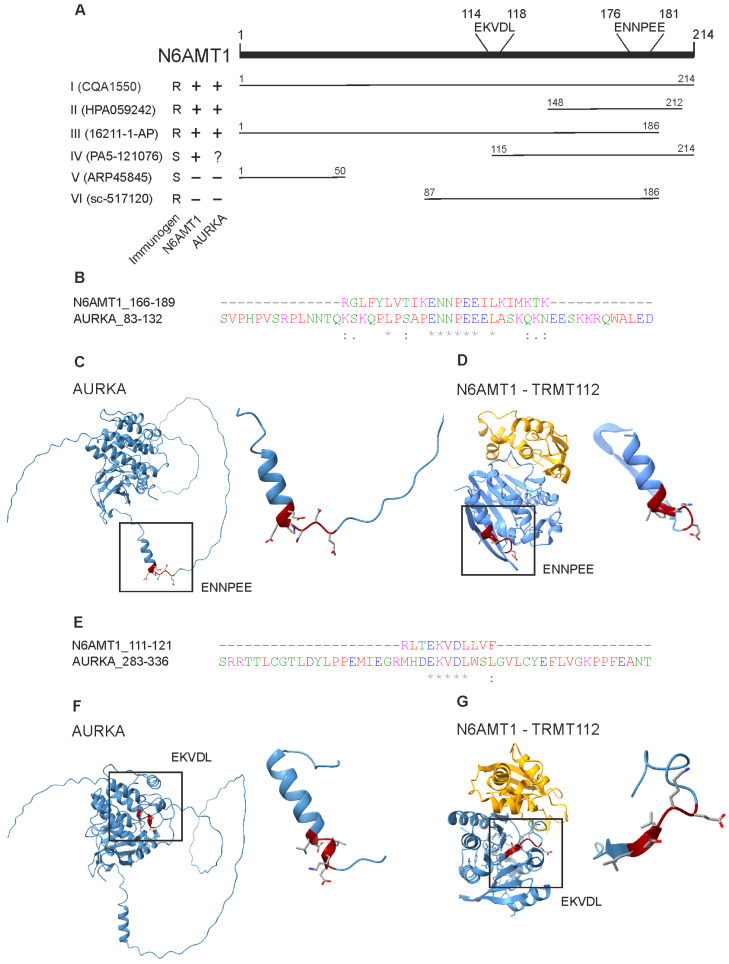
N6AMT1 and Aurora kinase A share motifs ENNPEE and EKVDL. (**A**) Schematic presentation of N6AMT1 protein with the localisation of EKVDL and ENNPEE motifs. Analysed antibodies immunogen range and production recombinant (R) or synthetic (S) are indicated. The antibody specificity to N6AMT1 and Aurora kinase A is shown as positive (+), negative (−), or not clear (?). (**B**) N6AMT1 and AURKA share motif ENNPEE. An asterisk (*) indicates positions which have a single, fully conserved residue. A colon (:) indicates conservation between groups of strongly similar properties – scoring > 0.5 in the Gonnet PAM 250 matrix. A period (.) indicates conservation between groups of weakly similar properties – scoring =< 0.5 in the Gonnet PAM 250 matrix.(**C**) Predicted AlphaFold structure of Aurora kinase A (AF_AFO014965F1) with motif ENNPEE. (**D**) Crystal structure of N6AMT1-TRMT112 complex with motif ENNPEE (PDB accession 6KMR) (**E**) N6AMT1 and AURKA share motif EKVDL. An asterisk (*) indicates positions which have a single, fully conserved residue. A : (colon) indicates conservation between groups of strongly similar properties – scoring > 0.5 in the Gonnet PAM 250 matrix. (F) Predicted AlphaFold structure of Aurora kinase A (AF_AFO014965F1) with motif EKVDL. (G) Crystal structure of N6AMT1-TRMT112 complex with motif EKVDL (PDB accession 6KMR).

**Table 1 antibodies-13-00033-t001:** Antibodies against N6AMT1.

Nr	Anti-N6AMT1	Host	Clonality	Immunogen	References
I	CQA1550; ABIN2966707; orb341253	r	Poly	1–214, R	[12]
II	HPA059242; PA5-66242	r	Poly	148–212, R	[19]
III	16211-1-AP	r	poly	1–186, R	[25,26]
IV	PA5-121076	r	poly	115–214, S	[9]
V *	PA5-42782; ARP45845_P050; orb579590; LS-C111069 ABIN2782381	r	poly	1–50, S	[27]
VI	sc-517120; ABIN565408	m	mono	87–186, R	n/a
VII	#27630	n/a	n/a	n/a	[8,10,11,28]
VIII	Discontinued—sc-83304	r	poly	n/a	[29]
VIV *	A7201	r	n/a	1–186, S	n/a
X *	STJ29281; NBP3-03312; ABIN6144314	r	poly	1–186, R	n/a
XI	ab238897; orb352915; ABIN7154967	r	poly	1–186, R	n/a
XII	abx005435; orb247851; GTX32649	r	poly	1–186, R	n/a
XIII	abx030019; orb165187; ABIN1538845	r	poly	1–30, S	n/a
XIV	abx235532; SH-A13857	r	poly	1–214, R	n/a
XV	ABIN1713839	r	poly	1–100, S	n/a
XVI	ABIN949232	r	poly	1–186, R	n/a
XVII	ABIN2752353	m	poly	1–186, R	n/a
XVIII	ABIN7116668	r	poly	1–214, R	n/a
XIX	abx320643	r	poly	1–186, R	n/a
XX	abx112962	r	poly	1–214, R	n/a
XXI	orb649232	r	poly	1–214, R	n/a
XXII	A10854	r	poly	n/a	n/a

R—recombinant; S—synthetic; r—rabbit; m—mouse; * Reference images are the same for these antibodies, but the immunogen is different.

## Data Availability

All relevant data are within the manuscript and its Appendix A.

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
