# Peer review of "Cross-Reactivity of N6AMT1 Antibodies with Aurora Kinase A: An Example of Antibody-Specific Non-Specificity"

_2073-4468, 2024, doi:10.3390/antib13020033_

Round 1

Reviewer 1 Report

Comments and Suggestions for Authors

The manuscript compared the specificity of several commercial N6AMT1 antibodies for proteins present in U2OS cells (wild type and N6AMT1 knockout cell line) using different methodologies. It is interesting that antibodies produced by immunogens with different sequences of N6AMT1 presented different characteristics and cross-reactivities with different proteins. The present work showed the importance of material selection and confident specifications provided by manufacturers since these parameters should influence in results and data reproducibility. Original immunoblotting images should be available for all articles to be able to visualize results without edition done by authors. 

Overall, the manuscript is well written and results are shown in the clear way.

Some comments are presented below:

- Lane 22: The first sentence “Antibody is one of the main tools of life science research.” It is very succinct and it is recommended an explanation about antibodies as one of the main tools of life science research.

- Lanes 33-35: Please give details about conflicting results concerning references 13-15. Is it related to antibodies used in these works?

- Lanes 45-48: Has the cross-reactivity of N6AMT1 antibodies with Aurora kinase A never been verified before? This information should be confirmed and added to the text.

- Figs. 2 and 3: Both figures could be combined and showed as one figure since similar patterns were observed in both mitotic U2OS and U2OS-ΔN6AMT1#1 cells. It is not necessary to show the same patterns in separate figures. This information should be clear in Figure caption.

Comments on the Quality of English Language

The overall English language is good and it is recommended a revision to improve.

Author Response

We are very thankful to the reviewer for reading our manuscript and giving us valuable feedback. We have addressed the raised questions and suggestions below.

Comment: Lane 22: The first sentence “Antibody is one of the main tools of life science research.” It is very succinct and it is recommended an explanation about antibodies as one of the main tools of life science research.

Response: We thank the reviewer for the comment. We have changed the sentence and included a sentence explaining antibody use in life science research (lanes 22-25).

 Lanes 22-25 “Primary antibodies are widely used tools in life science research. Antibodies have enabled life scientists to detect specific targets through various molecular biology methods, such as immunoblot, immunofluorescence, immunohistochemistry, immunoprecipitation, etc.”

Comment:  Lanes 33-35: Please give details about conflicting results concerning references 13-15. Is it related to antibodies used in these works?

Response: These results are not related to antibodies analysed in the current study. Some studies have shown that N6AMT1 is a 6mA DNA methyltransferase, while other studies have shown that N6AMT1 does not have 6mA DNA methyltransferase activity and based on its crystal structure, is not suitable for DNA methylation.

Comment: Lanes 45-48: Has the cross-reactivity of N6AMT1 antibodies with Aurora kinase A never been verified before? This information should be confirmed and added to the text.

Response: We thank the reviewer for the comment. To our knowledge, N6AMT1 antibody cross-reactivity with Aurora kinase A has not been reported in any studies prior. We have added this information to the main text (lane 51-52)

Lanes 51-52 “To our knowledge, prior studies have not detected N6AMT1 antibody cross-reactivity with Aurora kinase A.”

Comment: Figs. 2 and 3: Both figures could be combined and showed as one figure since similar patterns were observed in both mitotic U2OS and U2OS-ΔN6AMT1#1 cells. It is not necessary to show the same patterns in separate figures. This information should be clear in Figure caption.

Response: We agree with the reviewer’s comment. We have removed Figure 3 from the manuscript and added it as Supplementary Figure S2.

Comment: The overall English language is good and it is recommended a revision to improve.

Response: We thank the reviewer for the comment. We have carefully revised the manuscript to improve the English language.

Reviewer 2 Report

Comments and Suggestions for Authors

The article by Baiba Brūmele, et. al. entitled " Cross-reactivity of N6AMT1 antibodies with Aurora kinase A: an example of antibody-specific unspecificity" is a study regarding antibody specificity. The authors encountered issues related to cross-reactivity of N6AAMT1 antibodies in their experiments. They analyzed commercially available antibodies to solve the issues. Several antibodies against N6AMT1 showed cross-reactivity with the mitosis-related protein Aurora kinase A. They revealed that N6AMT1 and AURKA share the protein motif ENNPEE and N6AMT1 antibodies can recognize the motif in both proteins.

In general, the article is written clearly and presents interesting data, even if the tables are not well represented. This article would be of interest to scientists who are focusing on the study of antibodies.

I have the following comments and concern.

1.      Page 13, line 393, Do you have any comments regarding antibody II recognizing AURKA but antibody IV not?

2.      Can antibodies against AURKA recognize N6AMT1?

Minor points

There are careless typographical errors throughout the manuscript, even in the figure legends. I would recommend the authors to proofread the manuscript carefully. Below are some examples.

Page 4, Table 1, Does not cross pages.

Page 8, line 248-251 and 262-265, It seems to be same paragraph. Please delete one of them.

Page 9, Figure 5, line 286, “N6AMT1 (lane 2) and TRMT112 (lane 3)” should be “N6AMT1 (lane 3) and TRMT112 (lane 2).

Author Response

We are very thankful to the reviewer for reading our manuscript and giving us valuable feedback. We have addressed the raised questions and suggestions below.

Comment: Page 13, line 393, Do you have any comments regarding antibody II recognizing AURKA but antibody IV not?

Response:  We thank the reviewer for the question. Antibody IV produced a very high background with multiple bands that were much stronger than the band corresponding to N6AMT1. There was no unspecific band corresponding to endogenous AURKA, but there were unspecific bands corresponding to AURKA recombinant protein sizes. Therefore, we could not conclusively exclude whether there is any cross-reaction in immunoblot with AURKA. Nonetheless, even if there were any affinity towards AURKA recombinant proteins, it would have been low as there were no visible changes in the unspecific band strength. We have changed the sentence in lines 391-394 to describe our conclusions more accurately.

Lines 391-394 “Meanwhile, N6AMT1 antibody IV was also raised against the N-terminal part of the protein from 115-214 and included a full-length ENNPEE motif and partial EKVDL motif. This antibody produced multiple unspecific bands in immunoblot but did not recognise AURKA recombinant protein in the immunoblot assay with high affinity.”

Comment: Can antibodies against AURKA recognize N6AMT1?

Response:  We thank the reviewer for the question. We have considered the option and we have controlled it with one AURKA antibody in separate immunoblot. The mouse monoclonal AURKA antibody sc-56881 does not recognise N6AMT1, there is no visible band at the 23 kDa size, nor does it recognise our recombinant EGFP tagged N6AMT1 protein.

Comment: There are careless typographical errors throughout the manuscript, even in the figure legends. I would recommend the authors to proofread the manuscript carefully. Below are some examples.

Response: We thank the reviewer for the comment. We have addressed the issues mentioned, and we have carefully proofread the manuscript multiple times to eliminate the remaining errors.

Comment: Page 4, Table 1, Does not cross pages.

Response:  We have formatted the table in a way that it does not cross pages anymore.

Comment: Page 8, line 248-251 and 262-265, It seems to be same paragraph. Please delete one of them.

Response:  Repeating paragraph is deleted

Comment: Page 9, Figure 5, line 286, “N6AMT1 (lane 2) and TRMT112 (lane 3)” should be “N6AMT1 (lane 3) and TRMT112 (lane 2).

Response:  The lane numbers in Fig 5 are fixed.